# A Natural History of Erectile Dysfunction in Elderly Men: A Population-Based, Twelve-Year Prospective Study

**DOI:** 10.3390/jcm11082146

**Published:** 2022-04-12

**Authors:** Jouko Saramies, Markku Koiranen, Juha Auvinen, Hannu Uusitalo, Esko Hussi, Sebastian Becker, Sirkka Keinänen-Kiukaanniemi, Jaakko Tuomilehto, Kadri Suija

**Affiliations:** 1South Karelia Social and Health Care District, 53130 Lappeenranta, Finland; hussiesko@gmail.com (E.H.); sebastian.becker@eksote.fi (S.B.); 2Center for Life Course Health Research, University of Oulu, 90014 Oulu, Finland; markku.koiranen@oulu.fi (M.K.); juha.auvinen@oulu.fi (J.A.); skk@sun3.oulu.fi (S.K.-K.); kadri.suija@ut.ee (K.S.); 3Medical Research Center, Oulu University Hospital, 90014 Oulu, Finland; 4Department of Ophthalmology, Faculty of Medicine and Health Technology, Tampere University, PL 100, 33014 Tampere, Finland; hannu.uusitalo@tuni.fi; 5Tays Eye Centre, Tampere University Hospital, 33014 Tampere, Finland; 6Healthcare and Social Services of Selänne, 98530 Pyhäjärvi, Finland; 7Public Health Promotion Unit, Finnish Institute for Health and Welfare, 00280 Helsinki, Finland; jaakko.tuomilehto@helsinki.fi; 8Diabetes Research Group, King Abdulaziz University, Jeddah 21589, Saudi Arabia; 9Institute of Family Medicine and Public Health, University of Tartu, 50411 Tartu, Estonia

**Keywords:** erectile dysfunction, follow-up studies, incidence, lifestyle, prevalence

## Abstract

There is a wide variation in the development and course of erectile dysfunction (ED) in men, which confirms the need for prospective studies. We conducted a cross-sectional analysis among the general male population at the baseline (*n* = 359) and in a follow-up survey (*n* = 218) 12 years later. The prospective 12-year study included 189 men. ED was assessed using the International Index of Erectile Function questionnaire. The mean age of the participants was 62.0 years at the baseline, while at the 12-year follow-up it was 71.6 years. The crude prevalence of ED was 61.6% at the baseline and 78.9% at the follow-up, and the prevalence tended to increase with age. All of the men aged 75 years or more had at least mild ED. The incidence of ED in every thousand person years was 53.5. A total of 54.5% of the men experienced ED progression, while 39.2% reported no changes in erectile function, and 6.3% experienced ED regression during the 12-year study. The likelihood of ED progression was higher in the older compared with younger age group (odds ratio, OR 5.2 (95% CI: 1.1–26.2)), and the likelihood of ED regression was lower among men with increased depression symptoms (OR 0.3 (95% CI: 0.1–0.6)) and among men with a decreased interest in their sexual life (OR 0.1 (95% CI: 0.0–0.6)). Lifestyle factors such as the consumption of alcohol and smoking were not significantly associated with ED.

## 1. Purpose

The aim of this study was to investigate erectile function using a twelve-year follow-up study in order to uncover the prevalence and incidence of ED, and to be able to determine the possible association of factors—such as socio-demographic, lifestyle, or mental health aspects—with ED in elderly men in the general population.

## 2. Introduction

Erectile dysfunction (ED) is a common health problem in men; it is associated with normal aging as well as illness. In a large, population-based survey of 28,691 men, those who were aged between 70 and 75 years had a fourteen-fold relative risk of ED when compared to men who were aged between 20 and 29 years [1]. Various chronic diseases—such as heart disease [2], hypertension [3], diabetes [4], depression [5], obesity [6], and dyslipidemia [7]—have been linked to ED. Moreover, certain lifestyle factors—such as smoking [8] and low levels of physical activity [9]—are common in men who experience ED. In addition, for some drugs—such as thiazide diuretics, beta-blockers, spironolactone, serotonin inhibitors, and so on—a potential side effect is ED [10].

While the prevalence of ED and risk factors that can be linked with it are relatively well studied, there are fewer publications which cover incidences of ED in the general population. The incidence of ED varies between 26 and 99 cases in every 1000 person years [11], being influenced by factors such as the definition of ED, the man’s age, and the length of the follow-up period. The ‘Massachusetts male aging study’ (MMAS) and the ‘Men’s attitudes to life events and sexuality’ (MALES) study followed selected men for a period of up to nine years [12]. The results from the longest follow-up study showed that approximately 30% of men experienced ED progression and 30% experienced ED remission during the follow-up period [13]. The risk factors in terms of progression included age, a high body mass index (BMI), and current smoking at the baseline. On the other hand, in a five-year prospective study of 466 men with diabetes and ED, only 9% experienced ED remission, while those who were younger had a shorter history of diabetes, and their experience of ED had psychological, rather than somatic, causes [14].

There is still little information available on how those factors which have been found to be important during cross-sectional studies tend to contribute to the potential severity of ED, or on the ways in which the duration of any follow-up serves to influence prevalence and incidence estimates.

## 3. Materials and Methods

Based on information supplied by the population registry center, we sent out postal invites to everyone within the 41–66 age group who was living in the rural community of Savitaipale, in eastern Finland, requesting that they participate in the study in the 1996–1999 period. The target population comprised 800 men. Ten years later, in 2007–2008, we carried out the first erectile function assessment of 433 of the men, of whom 359 (83%) completed the Index of Erectile Function (IIEF-5) questionnaire. This was the baseline for erectile dysfunction assessment. In 2018–2019, a total of 305 men participated in the twelve-year follow-up study, of whom 218 (71%) completed the IIEF-5. Between the baseline and the follow-up, 121 men died, such that the participation rate in the surviving men was 305 out of 312 men (98%). Altogether, a total of 189 men participated twice in the erectile function assessment, both at the baseline and follow-up.

Figure 1 presents the flowchart for the study group.

Both surveys in our study included questionnaires and clinical measurements. Weight, height, and blood pressure were measured in all of the participating men by a research nurse. Their BMI was calculated using the following formula: current weight (kg) divided by the square of the baseline height (m). Obesity, as defined using BMI, was ≥30 kg/m^2^.

The questionnaires in all of our surveys included socio-demographic details and information about health behavior and lifestyle such as smoking and alcohol consumption. Symptoms of depression were assessed by means of the Beck Depression Inventory (BDI-I); a score of <10 was considered to be the threshold for non-depression, while those who had a score of ≥10 were defined as having depressive symptoms [18]. Concerning smoking, a subject was defined as having ‘stopped smoking’ if he had indeed stopped smoking at least one year ago. The amount of alcohol which equaled a typical drink in our study was measured at twelve grams of alcohol, and we asked the men about their alcohol consumption during the preceding week.

We used a pen and paper format for all of the questionnaires. A quiet, isolated space was set aside in which the participants answered the questionnaire on an individual basis. The IIEF-5 form was included, along with the other study questionnaires.

The study design has previously been described in detail [19].

We compared the characteristics of the participants at the baseline and during the twelve-year follow-up according to their erectile function using the Pearson chi-squared statistics cross-sectional process, and we conducted the test for incidence difference. Statistical significance was taken as *p* < 0.05. Those men who reported having no sexual activity (*n* = 35 at the baseline figure and *n* = 53 at the twelve-year follow-up) also reported no confidence in their ability to maintain an erection based on the IIEF-5 (question number one), and were therefore included in the severe ED group. The odds ratios (ORs) with ninety-five percent confidence intervals (95% CI) in the prevalence of ED at the baseline figure and at the twelve-year follow-up were calculated for factors which were considered to be important in terms of the cross-sectional analysis, and the results were unadjusted and adjusted for potential confounding variables. A Sankey diagram was used to describe the changes in ED across the cohort from the baseline to the twelve-year follow-up point. We calculated the follow-up time and the incidence for every man and divided the number of cases by the person years, which were scaled to one thousand person years.

The factors that influenced the course of ED were evaluated in order to be able to determine those associated with the progression or regression of ED.

All of the statistical analyses were carried out using SAS Proprietary Software (TS1M5) 9.4 (SAS Institute Inc., Cary, NC, USA) and OriginPro 2019b (OriginLab®, Northampton, MA, USA).

The study was conducted according to the Declaration of Helsinki, and was approved by the Ethics Review Board of the South Karelia Hospital District. The Research Ethics Committee of the University of Helsinki approved the research (HUS/2203/2018).

## 4. Results

Figure 1 presents the flowchart for the study group.

Table 1 presents the general characteristics of the men who participated in the IIEF-5 study in the baseline survey, in the twelve-year follow-up, and in both of the surveys.

At the baseline, the mean age of the men who responded was 62.0 years, and it was 65.8 years for those males who did not respond to the study call. The means differed significantly (*p* < 0.001). At the twelve-year follow-up point, the mean ages were 71.6 and 74.4 years (*p* < 0.001). For the 189 men who participated in both of the surveys, the mean age was 60.1 years at the baseline and 71.3 years at the twelve-year follow-up point. Higher education and being of a normal weight or overweight reduced the incidence of ED. Social class, marital status, smoking, the consumption of alcohol, interest in one’s sexual life, and depressive symptoms did not affect the incidence of ED.

The prevalence of smoking, levels of alcohol use, and degrees of obesity, as well as any interest in a sex life—based on BDI-I question number 21—were all seen to decrease during the study period.

Figure 2a,b presents the prevalence of ED subgroups by age at the baseline and at the twelve-year follow-up. The prevalence of ED increased in all of the age groups during the follow-up.

The crude prevalence of ED was 61.6% at the baseline and 78.9% at the twelve-year follow-up. The prevalence of ED increased with age. Out of the youngest age group (aged between 50 and 54 years), a total of 38.6% had ED, while all of the men who were older than 75 years had ED. The severity of ED also increased with age, with 9% of the men aged 50–54 years having severe ED and 65% aged 80–86 years having severe ED.

Table 2 presents the logistic model of ED for those factors which were significantly correlated with ED within the cross-sectional analysis (which is presented in Table 1). Concerning age, the OR of ED in the older age group of 70 years or more was 7.9 (95% CI: 1.4–43.6) at the baseline and 3.5 (95% CI: 1.1–11.3) at the follow-up of the subjects who participated in both surveys. Additionally, being single, divorced, or widowed, and having reduced or no interest in a sexual life were associated with an increased prevalence of ED: the ORs were 3.0 (95% CI: 1.2–7.5) and 4.0 (95% CI: 1.9–8.4), respectively, at the baseline, and 4.1 (95% CI: 1.1–14.4) and 3.3 (95% CI: 1.3–8.1), respectively, at the follow-up.

Table 3 and Figure 3 presents the changes in erectile function during the study period. During the twelve-year study period, there were 54 new ED cases, which put the incidence of ED at 60.7%, and the incidence of ED per thousand person years was 53.5.

Within the cohort of 189 men who were followed closely for a period of twelve years, we found that 103 (54.5%) of them experienced ED progression, while 74 (39.2%) reported no change in erectile function, and 12 (6.3%) experienced ED regression. Mild erection disfunction in seven men returned to normal. Of the eighty-nine men who still reported normal erectile function at the baseline, a total of thirty-five (39.3%) also experienced normal erectile function after the twelve-year period.

Table 4 presents the factors that were considered to be important in terms of the progression and regression of ED. Those factors that were regarded as being statistically significant in terms of improved erectile function during the twelve-year period were age, BDI score, and interest in a sexual life. The probability that erectile function remained the same or improved was smaller among 65–69-year-old men, OR 0.1 (95% CI: 0.0, 0.9)); among those with an increased BDI score, OR 0.3 (95% CI: 0.1–0.6)); and among those with a decreased interest in a sexual life, OR 0.1 (95% CI: 0.0, 0.6). The probability of erection persistence was higher among men with an increased BDI score (the OR of ED was 1.9 (1.1, 3.5)), having basic education (2.6 (95% CI: 1.4, 4.8)) and among single, divorced, or widowed men (2.5 (95% CI: 1.2, 5.2)). Older age (70–75 years) was regarded as being statistically significant in terms of the progression of ED (erectile function decreased) (the OR of ED was 5.2 (95% CI: 1.1, 26.2)).

## 5. Discussion

We found that more than half of the men at the baseline (61.6%) and most of them (78.9%) at the twelve-year follow-up had some level of ED, with the incidence of ED per thousand person years being 53.5. Similar figures for ED prevalence have also been reported by other researchers who used the same instrument to assess ED in men of similar ages [20,21]. The incidence numbers vary more in various studies (19.2–65.6), which quite often used different instruments to those used in this study (such as only employing one or two questions) and had a shorter study period than our study did [22].

The number of men in our study who reported ED tended to increase with age; at the age of seventy-five or above, all of the men had experienced at least mild levels of ED. Age was also statistically significant when associated with ED progression during the twelve-year study. Age is the most significant factor that affects ED, as reported by almost all of the studies in this field [20]. ED caused by aging is probably due to various hormonal and structural changes in the body. Even though it is already well known that erectile function diminishes during the aging process, there is no consensus about whether there is a certain age at which it becomes physiologically ‘normal’ to experience ED.

We found that ED was associated with interest sex abstinence. The number of men who reported that they experience no sexual activity increased during the twelve-year follow-up, from 9.8% to 24.3%. Obviously, this issue is bidirectional; having erectile problems would tend to diminish any interest in a sex life and vice versa, which may result in experiencing no sexual activity at all. On the other hand, little interest in a sex life may also be one of the symptoms of depression, with depressive disorders also being one of the known risk factors for ED in elderly men [23]. Amongst this study’s population, men who had symptoms of depression had greater degrees of ED compared to men who experienced no symptoms of depression. Changes in symptoms of depression (Table 4) were also shown to be significantly associated with ED during the twelve-year study. ED influences the quality of life of those who have it, along with the mental wellbeing of those who have it [24]. Therefore, the ability to be able to take into account possible symptoms of depression is something which remains very relevant in the management of erectile problems.

We found that being single, divorced, or widowed tended to be associated with ED when compared to those men who were married or who were cohabiting. Unfortunately, there was no precise question in the study regarding a sexual partner or the number of times intercourse had been attempted. Therefore, we were unable to draw any direct inferences, although our results may indicate that ED is more common in men who do not have a consistent partner. Based on previous studies, we know that sexual activity diminishes with age, along with a rise in the prevalence of health problems and the potential for the loss of a partner [25,26]. On the other hand, it has been shown that regular intercourse may protect subjects against the development of ED [27]. Moreover, seeking help for ED is something that can be associated with stigma [28]. Therefore, proactively asking elderly men about their potential ED is important. It may positively help them in terms of their erectile function and sexual activity.

Interestingly, smoking and the consumption of alcohol were not significantly associated with ED in our study population, unlike the results shown in previous studies [8]. This could be because the men who reported that they did not consume alcohol at all, or that they had stopped smoking, had done so due to their health problems. It is unclear whether this could be happening due to the presence of ED. The recommendations not to smoke, not to drink alcohol, and to be physically active are the most common recommendations for patients who have ED, as well as for many other health complaints. The number of responses from male smokers declined the most, possibly due to higher mortality. On the other hand, ED in smokers within the cohort of men who participated in both surveys tended to double, from 34.3% to 68.6%, but, due to the small numbers of respondents involved, this was not statistically significant. The inability to find associations with lifestyle factors may also be related to an underestimation of alcohol consumption and smoking, which is typical in questionnaire surveys or when related to a small number of participants. On the other hand, ED risk factors are more commonly studied in a cross-sectional format, so it is possible that, during the long follow-up period of twelve years, the role of lifestyle factors tended to decrease. Obviously, these aspects need to be studied further.

During the twelve-year follow-up period, a total of 54.5% of the men in our study population reported that their erectile function levels had declined. On the other hand, 6.3% of the men in the study reported improved erectile function levels. Of the 89 men with normal erectile function at the baseline figure, a total of 39.3% still had no symptoms of ED after twelve years. Therefore, their erectile function levels had been maintained despite the ongoing aging process. This result agrees with findings from previous observational studies that indicated that the presence of ED is not a fixed status [12].

The main strength of our study is the long follow-up period of twelve years, which, to our knowledge, is the longest follow-up period that has been used in any ED study. Our data are based on a homogenous population-based cohort with a simultaneous investigation of erectile function and other parameters.

There were some limitations in our study. The definition of ED could affect the evaluation of the incidence and prevalence of ED. ED can be estimated through the use of self-administered questionnaires along with the study of symptoms, and through objective measurements. Objective tests are used only in selected patients in urology/andrology clinics. The IIEF-5 is subjective, using attributes such as ‘sometimes’ or ‘most times’. However, questionnaires such as the IIEF-5 are widely used in epidemiological studies, as they are non-invasive, easy to use, and are most often standardized and validated; they also significantly optimize levels of ED management [29]. Well-known problems with self-reported data in terms of recall bias and other biases may have influenced responses, in particular in relation to ED and potential confounders (such as alcohol, smoking, physical activity, etc.). The low response rate to the IIEF-5 questionnaire may have had an impact on the relevance of the results. The men who did not respond to the questionnaire may have assumed that they were no longer affected by the potency assessment, and may have had a higher rate of erectile dysfunction than those who responded. The proportion of valid answers decreased with age, with 92% of men aged between 50 and 59 years responding, while 83% of those aged between 60 and 69 years responded, and 68% of those aged seventy years or above responded. Additionally, the response levels may have been affected by the fact that the research nurses themselves lived in the same village as the men.

## 6. Conclusions

ED is common in elderly men. The long follow-up period of twelve years used in this study made it possible to gather new depths of information about changes in ED which are related to the aging process. All of the men who were aged above seventy-five had at least mild symptoms of ED in our study population. Even though ED is common and increases in prevalence with age, it is not a fixed state. Higher education and being of normal weight or overweight were associated with a lower incidence of ED.

## Figures and Tables

**Figure 1 jcm-11-02146-f001:**
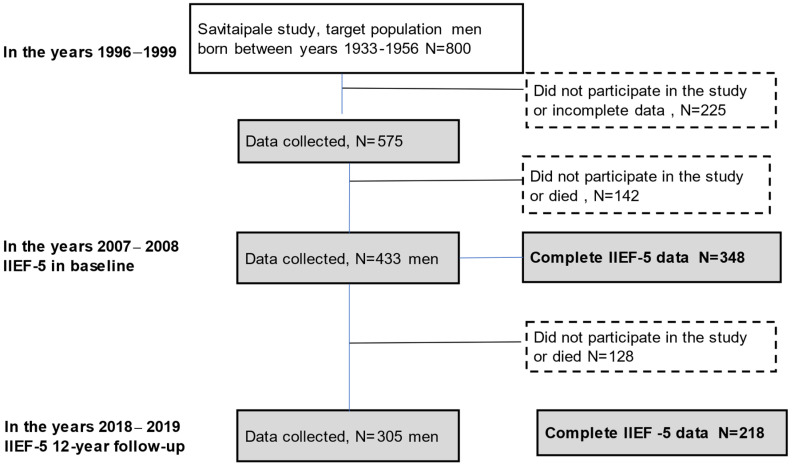
The flowchart for the study groupWe used the IIEF-5 to assess erectile function [15,16]. The IIEF-5 is a shorter version of the original IIEF questionnaire which comprises four questions from the sexual function domain of the original version and one question from the intercourse satisfaction domain. The IIEF was developed and then cross-culturally and linguistically validated in 1996–1997, while also being psychometrically tested. The IIEF-5 was originally designed to assess erectile function over the preceding four weeks. A period of six months was adopted for the standard assessment phase, following the recommendation of the National Institutes of Health [17]. Each question in the IIEF-5 is graded on a five-point scale, while four questions also include a score of zero, which indicates a totally lack of sexual activity. Based on the IIEF-5 scores, erectile function can be classified into five categories: severe ED (scoring 5–7 points), moderate ED (8–11 points), mild to moderate ED (12–16 points), mild ED (17–21 points), and no ED (22–25 points).

**Figure 2 jcm-11-02146-f002:**
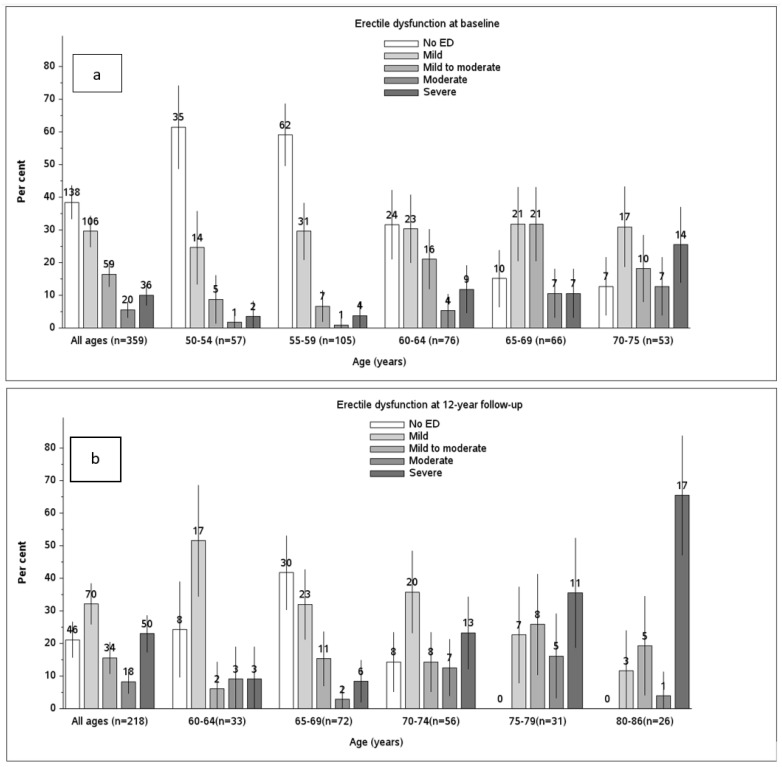
The proportion of men who reported a normal erection ability, or mild, mild to moderate, moderate, or severe ED at the baseline (**a**) and at the twelve-year follow-up (**b**). The figures shown above the bars illustrate the number of men, and the vertical lines show the 95% confidence intervals for the prevalence of ED.

**Figure 3 jcm-11-02146-f003:**
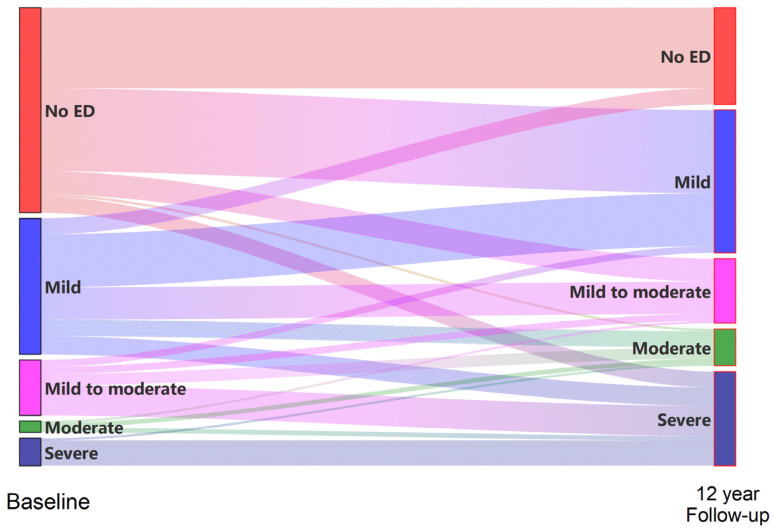
Changes in erectile function flow and changes according to the IIEF-5 during the 12-year follow-up period.

**Table 1 jcm-11-02146-t001:** Distribution figures for specific socio-demographic and lifestyle variables, along with the frequency and prevalence (%) of erectile dysfunction (ED) in the categories for these variables in the baseline and twelve-year follow-up studies.

	Men Who Participated in the Baseline Study, *N* = 359		Men Who Participated in the Follow-Up Study, *N* = 218		Men Who Participated Both in the Baseline and Follow-Up Studies, *N* = 189			Incidence ** *N* (%) *** *p*
	All	ED Prevalence	All	ED Prevalence	All	ED Prevalence at Baseline	ED Prevalence at Follow-up	
	*N*	%	*N*	%	*N*	*N* (%)	*N* (%)	
**Age group**		*** *p* ≤ 0.001**		*** *p* < 0.001**		*** *p* < 0.001**	*** *p* ≤ 0.001**	
50–54	57	38.6			46	16 (34.8)	33 (71.7)	17 (56.7) 0.581
55–59	105	41.0			60	24 (40.0)	36 (60.0)	19 (52.8) 0.209
60–64	76	68.4	33	75.8	42	25 (59.5)	37 (88.1)	12 (70.6) 0.352
65–69	66	84.8	72	58.3	25	21 (84.0)	25 (100)	4 (100.0) 0.099
70–74	55	87.3	56	85.7	16	14 (87.5)	16 (100)	2 (100.0) 0.250
75–79			31	100				
80–86			26	100				
Total	359	61.6	218	78.9	189	100(52.9)	147(77.8)	54 (60.7)

**Education**		*** *p* = 0.01**		*** *p* = 0.03**		*** *p* = 0.005**	*** *p* = 0.028**	
Higher	19	42.1	23	56.5	19	8 (42.1)	10 (52.6)	3 (27.3) 0.024
Vocational	74	41.9	77	77.9	69	27 (39.1)	52 (75.4)	27 (64.3) 0.279
Basic	92	64.1	94	81.9	82	53 (64.6)	67 (81.7)	18 (62.1) 0.631
Total	185	53.0	194	77.3	170	88 (51.8)	129 (75.9)	48 (58.5)
**Social class**		*** *p* = 0.81**		*** *p* = 0.38**		*** *p* = 0.78**	*** *p* = 0.42**	
Upper	55	60.0	60	75.0	49	28 (57.1)	36 (73.5)	11 (52.4) 0.491
Upper middle	18	50.0	20	65.0	18	9 (50.0)	12(66.7)	3 (33.3) 0.100
Middle	18	44.4	20	75.0	17	7 (41.2)	12 (70.6)	6 (60.0) 0.921
Working	56	53.6	56	80.4	52	27 (51.9)	42 (80.8)	17 (68.0) 0.233
Other	14	57	14	92.9	13	8 (61.5)	12 (92.3)	4 (80.0) 0.313
Total	161	54.)	170	77.1	149 (100)	79 (53.0)	114 (76.5)	
**Marital status**		*** *p* = 0.03**		*** *p* = 0.16**		*** *p* = 0.053**	*** *p* = 0.044**	
Married or cohabiting	271	58.3	160	75.6	150	74 (49.3)	112 (74.7)	45 (59.2) 0.494
Single, divorced, or widowed	83	71.1	42	85.7	39	26 (66.7)	35 (89.7)	9 (69.2) 0.494
Total	354	61.3	202	77.7	189	100 (52.9)	147 (77.8)	54 (60.7)
**Smoking**		*** *p* = 0.069**		*** *p* = 0.90**		*** *p* = 0.040**	*** *p* = 0.33**	
Never smoked	142	64.1	84	81	84	50 (59.5)	68 (81.0)	19 (55.9) 0.467
Stopped smoking	129	65.1	92	78.3	64	35 (54.7)	50 (78.1)	19 (65.5) 0.516
Current smoker	76	50.0	19	78.9	35	12 (34.3)	24 (68.6)	13 (56.5) 0.636
Total	347	61.4	195	79.5	183	97 (53.0)	142 (77.6)	54 (60.7)

**Weekly alcohol consumption (g)**		*** *p* = 0.18**		*** *p* = 0.44**		*** *p* = 0.10**	*** *p* = 0.52**	
No alcohol consumption	43	72.1	33	84.8	23 (14.6)	16 (69.6)	19 (82.6)	3 (42.9) 0.339
<168 g	210	57.1	116	75.0	120 (75.9)	56 (46.7)	92 (76.7)	39 (60.9) 0.635
≥168 g	32	62.5	14	71.4	15 (9.5)	9 (60.0)	10 (66.7)	4 (66.7) 0.719
Total	285	60.0	163	76.7	158 (100)	81 (51.3)	121 (76.6)	46 (59.7)
**Body Mass Index**		*** *p* = 0.47**		*** *p* = 0.07**		*** *p* = 0.96**	*** *p* = 0.03**	
<30 kg/m^2^	289	60.6	185	76.8	159	84 (52.8)	119 (74.8)	42 (56.0) 0.037
≥30 kg/m^2^	69	65.2	33	90.9	30	16 (53.3)	28 (93.3)	12 (85.7) 0.037
Total	358	61.5	218	78.9	189	100 (52.9)	147 (77.8)	54 (60.7)
**Interest in sexual life**		*** *p* < 0.001**		*** *p* < 0.001**		*** *p* < 0.001**	*** *p* = 0.095**	
As before	180	43.9	92	66.3	104	38 (36.5)	75 (72.1)	38 (57.6) 0.311
Reduced	171	78.4	108	87.0	82	59 (72.0)	69 (84.1)	16 (69.6) 0.311
Not at all	7	100	11	100	3	3 (100)	3 (100)	0 (0.0)
Total	358	61.5	211	78.7	189	100 (52.9)	147 (77.8)	54 (60.7)
**BDI-I score**		*** *p* = 0.002**		*** *p* = 0.015**		*** *p* = 0.016**	*** *p* = 0.339**	
<10	332	59.3	198	76.8	179	91 (50.8)	138 (77.1)	51 (60.7) 0.975
≥10	27	88.9	20	100	10	9 (90.0)	9 (90.0)	3 (60.0) 0.975
Total	359	61.6	218	78.9	189	100 (52.9)	147 (77.8)	54 (60.7)

* *p*-values indicates whether the prevalence of ED is equal in each variable category(chi square test). ** The prevalence and incidence differ because the erectile function in seven men returned to normal. *** *p*-value indicates whether the observed incidence of a particular class is the same as the expected (total) incidence.

**Table 2 jcm-11-02146-t002:** Logistic model with erectile dysfunction as the outcome, and statistically significant variables from Table 1 as risk factors. The unadjusted results, those with other variables’ adjusted odds ratios (OR), and their 95 percent confidence intervals (95% CI) are presented.

Variable	All Men				Men Participated to Both Baseline and Follow-Up			
	Baseline (*N* = 359)		Follow-Up (*N* = 172)		Baseline (*N* = 189)		Follow-Up (*N* = 189)	
Category	Unadjusted OR (95% CI)	Adjusted OR (95% CI)	Unadjusted OR (95% CI)	Adjusted OR (95% CI)	Unadjusted OR (95% CI)	Adjusted OR (95% CI)	Unadjusted OR (95% CI)	Adjusted OR (95% CI)
**Age (years)**								
50–54	ref	ref			ref	ref		
55–59	1.1 (0.6,2.1)	0.9 (0.4,1.9)			1.3 (0.6,2.8)	1.0 (0.4,2.5)		
60–64	3.4 (1.7,7.1)	2.5 (1.1,5.6)	ref	ref	2.8 (1.2,6.5)	2.6 (1.0,7.1)	ref	ref
65–69	8.9 (3.8,21.0)	6.1 (2.4,15.8)	0.4 (0.2,1.1)	0.4 (0.1,1.2)	9.8 (2.9,33.7)	10 (2.6,40.1)	0.4 (0.2,1.2)	0.4 (0.1,1.2)
70 and over	11.0 (4.2,28.4)	6.3 (2.2,18.1)	4.2 (1.4,12.3)	3.8 (1.2,12.1)	13.0 (2.6,65.1)	7.9 (1.4,43.6)	3.7 (1.2,10.8)	3.5 (1.1,11.3)
**Education**								
Higher or vocational	ref	ref	ref	ref	ref	ref	ref	ref
Basic	2.5 (1.4,4.5)	2.7 (1.4,5.4)	1.7 (0.8,3.3)	1.5 (0.7,3.6)	2.8 (1.5,5.2)	3.0 (1.4,6.3)	1.9 (0.9,3.9)	1.7 (0.7,4.0)
**Marital status**								
Married or cohabiting	ref	ref	ref	ref	ref	ref	ref	ref
Single, divorced, or widowed	1.8 (1.0,3.0)	2.1 (1.1,3.9)	1.9 (0.8,4.9)	3.0 (0.9,9.7)	2.1 (1.0,4.3)	3.0 (1.2,7.5)	3.0 (1.0,8.9)	4.1 (1.1,14.4)
Body mass index								
<30 kg/m^2^	ref	ref	ref	ref	ref	ref	ref	ref
≥30 kg/m^2^	1.5 (0.8,2.8)	1.1 (0.5,2.4)	3.0 (0.9,10.4)	3.6 (0.7,18.6)	0.7 (0.3,1.8)	0.7 (0.2,2.4)	4.1 (0.9,18.1)	3.7 (0.7,19.5)
Interest in sexual life								
As before	ref	ref	ref	ref	ref	ref	ref	ref
Reduced or not at all	4.9 (3.1,7.8)	3.3 (1.9,5.7)	3.8 (1.9,7.7)	2.8 (1.2,6.8)	4.7 (2.5,8.7)	4.0 (1.9,8.4)	4.1 (1.9,8.6)	3.3 (1.3,8.1)
Beck’s BDI-I score								
<10	ref	ref	ref	ref	ref	ref	ref	ref
≥10	3.7 (1.8,7.7)	2.4 (1.1,5.6)	5.4 (1.3,23.5)	2.5 (0.5,12.8)	3.9 (1.4,11.1)	3.4 (1.0,11.3)	5.1 (1.2,22.4)	2.3 (0.4,12.3)

**Table 3 jcm-11-02146-t003:** Prevalence and changes in erectile dysfunction from the baseline to the 12-year follow-up.

	ED at the Baseline					
	No ED	Mild	Mild to Moderate	Moderate	Severe	Total
ED at 12-Year Follow-Up	*N* (%)	*N* (%)	*N* (%)	*N* (%)	*N* (%)	*N* (%)
No ED	35 (39.3)	7 (11.9)				42 (22.2)
Mild	36 (40.4)	23 (39.0)	3 (12.5)			62 (32.8)
Mild to moderate	10 (11.2)	14 (23.7)	3 (12.5)	1 (20.0)		28 (14.8)
Moderate	1 (1.1)	7 (11.9)	5 (20.8)	2 (40.0)	1 (8.3)	16 (8.5)
Severe	7 (7.9)	8 (13.6)	13 (54.2)	2 (40.0)	11 (91.7)	41 (21.7)
Total	89 (100)	59 (100)	24 (100)	5 (100)	12 (100)	189 (100)

**Table 4 jcm-11-02146-t004:** Factors which were important in terms of the progression and regression of ED.

	Erectile Function Remained Same or Improved			Erectile Function Decreased			Erectile Dysfunction Persistence		
	*N*	%	OR (95% CI)	*N*	%	OR (95% CI)	*N*	%	OR (95% CI)
**Baseline age**		***p* ≤ 0.001**			***p* = 0.009**			***p* ≤ 0.001**	
50–54	46	28.3	ref	45	53.3	ref	46	34.8	ref
55–59	60	41.7	1.8 (0.8,4.1)	59	44.1	0.7 (0.3,1.5)	60	28.3	0.7 (0.3,1.7)
60–64	42	19	0.6 (0.2,1.6)	37	64.9	1.6 (0.7,3.9)	42	59.5	2.8 (1.2,6.5)
65–69	25	4	0.1 (0.0,0.9)	22	77.3	3.0 (0.9,9.5)	25	84	9.8 (2.9,33.7)
70–75	16	0	n.a.	14	85.7	5.2 (1.1,26.2)	16	87.5	13.1 (2.6,65.1)
**Education**		***p* = 0.073**			***p* = 0.568**			***p* = 0.002**	
Higher or vocational	88	33	ref	81	55.6	ref	88	36.4	ref
Basic	82	20.7	0.5 (0.3,1.1)	80	60	1.2 (0.6,2.2)	82	59.8	2.6 (1.4,4.8)
**Social class**		***p* = 0.280**			***p* = 0.285**			***p* = 0.699**	
Upper	49	30.6	ref	46	54.3	ref	49	51	ref
Upper middle	18	38.9	1.4 (0.5,4.4)	17	52.9	0.9 (0.3,2.9)	18	50	1.0 (0.3,2.8)
Middle	17	29.4	0.9 (0.3,3.2)	16	50	0.8 (0.3,2.6)	17	35.3	0.5 (0.2,1.6)
Working	52	21.2	0.6 (0.2,1.5)	50	56	1.1 (0.5,2.4)	52	48.1	0.9 (0.4,1.9)
Other	13	7.7	0.2 (0.0,1.6)	10	90	7.6 (0.9,64.6)	13	61.5	1.5 (0.4,5.4)
**Marital status**		***p* = 0.051**			***p* = 0.365**			***p* = 0.014**	
Married or cohabiting	150	28	ref	142	59.9	ref	150	44.7	ref
Single, divorced, or widowed	39	12.8	0.4 (0.1,1.0)	35	51.4	0.7 (0.3,1.5)	39	66.7	2.5 (1.2,5.2)
**Change in Smoking**		***p* = 0.476**			***p* = 0.668**			***p* = 0.136**	
Nonsmoker at baseline–nonsmoker at follow-up	133	23.3	ref	125	59.2	ref	133	51.9	ref
Smoker at baseline–nonsmoker at follow-up	13	38.5	2.1 (0.6,6.7)	11	45.5	0.6 (0.2,2.0)	13	38.5	0.6 (0.2,1.9)
Smoker at baseline–smoker at follow-up	15	26.7	1.2 (0.4,4.0)	15	60	1.0 (0.3,3.1)	15	26.7	0.3 (0.1,1.1)
**Change in alcohol use**		***p* = 0.627**			***p* = 0.774**			***p* = 0.038**	
No alcohol at all	16	37.5	ref	13	53.8	ref	16	75	ref
Same or decreased	75	26.7	0.6 (0.2,1.9)	72	61.1	1.3 (0.4,4.4)	75	40	0.2 (0.1,0.8)
Increased	51	25.5	0.6 (0.2,1.9)	47	55.3	1.1 (0.3,3.6)	51	45.1	0.3 (0.1,1.0)
**Change in BMI**		***p* = 0.328**			***p* = 0.791**			***p* = 0.329**	
BMI same or decreased	105	27.6	ref	96	57.3	ref	105	52.4	ref
MI increased	84	21.4	0.7 (0.4,1.4)	81	59.3	1.1 (0.6,2.0)	84	45.2	0.8 (0.4,1.3)
**Change in sexual interest**		***p* ≤ 0.001**			***p* ≤ 0.001**			***p* = 0.025**	
Increased	21	28.6	ref	20	65	ref	21	61.9	ref
No change	105	35.2	1.4 (0.5,3.8)	97	46.4	0.5 (0.2,1.3)	105	40	0.4 (0.2,1.1)
Decreased	57	5.3	0.1 (0.0,0.6)	55	76.4	1.7 (0.6,5.3)	57	59.6	0.9 (0.3,2.5)
**Change in Beck BDI-I**		***p* ≤ 0.001**			***p* = 0.118**			***p* = 0.024**	
Same or decreased	97	35.1	ref	93	52.7	ref	97	41.2	ref
Increased	92	14.1	0.3 (0.1,0.6)	84	64.3	1.6 (0.9,3.0)	92	57.6	1.9 (1.1,3.5)

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
