# Peer review of "A Natural History of Erectile Dysfunction in Elderly Men: A Population-Based, Twelve-Year Prospective Study"

_jcm, 2022, doi:10.3390/jcm11082146_

Round 1

Reviewer 1 Report

  1. The basic but critical issue is that some numbers/statistics in the manuscript text or Tables are not exact and without clear data resources. All about these must be rechecked and clarified. For example, (a)The crude prevalence of ED was 61.5% at the baseline figure and 78.8% 167 at the twelve-year follow-up, which is not the same with the results showed in the Table 1; (b) The number of men who reported that they experience no sexual activity was 53 of 228 at the point of the twelve-year follow-up. Where these numbers come from? etc.
  1. Where is the Table 4 ? All the tables are difficult to be read and must be revised.
  1. Please add more interpretation for the “Sankey graphic”(Figure 3).
  1. In the discussion, please explain how to estimate the incidence of ED for every thousand person-years being at 53.5.
  1. The authors mentioned that ED risk factors are more heavily studied in a cross-sectional format, so it is possible that, during the long follow-up period of twelve years, the role of lifestyle factors will tend to decrease. Based on the results in Table 1, the trend seems to increase for current smokers (50% to 78.9%) as well as alcohol consumption >=168g (62.5% to 71.4%). Please clarify it.
  1. Symptoms of depression also bore a significant association with a lack of progression in terms of ED during the subsequent twelve years. What that means?
  1. The conclusion not just focus on the age effect, the most important is that the authors should give a meaningful information regarding the “risk” or “protective” factors for ED.

Reviewer 2 Report

Dear Authors,

your study was well conducted and well written. Results are adequately described and meaningful, nevertheless it has some points to be further elucidated:

1) Please provide a statement regarding ethics also in the manuscript. Please also explain how the study was approved in 2018, but the first patients enrolled more than 12 years before. If a previous study was conducted, please report the previous study citation in the text and provide further explanation

2) In study limitation, please add at least a phrase regarding the low response rate of the questionnaire, which has an impact on the relevance of the results 

Round 2

Reviewer 1 Report

The authors have appropriately responded to most of reviewers' comments. The interpretations of their relevant findings are still needed to be improved with detail explanations.